# Adequacy of the 10 mg/kg Daily Dose of Antituberculosis Drug Isoniazid in Infants under 6 Months of Age

**DOI:** 10.3390/antibiotics12020272

**Published:** 2023-01-30

**Authors:** Maria Goretti López-Ramos, Joan Vinent, Rob Aarnoutse, Angela Colbers, Eneritz Velasco-Arnaiz, Loreto Martorell, Lola Falcón-Neyra, Olaf Neth, Luis Prieto, Sara Guillén, Fernando Baquero-Artigao, Ana Méndez-Echevarría, David Gómez-Pastrana, Ana Belén Jiménez, Rebeca Lahoz, José Tomás Ramos-Amador, Antoni Soriano-Arandes, Begoña Santiago, Rosa Farré, Clàudia Fortuny, Dolors Soy, Antoni Noguera-Julian

**Affiliations:** 1Pharmacy Department, Hospital Sant Joan de Déu, 08950 Barcelona, Spain; 2Department of Pharmacy, Radboud University Medical Center, Radboud Institute for Health Sciences, 6525 HB Nijmegen, The Netherlands; 3Malalties Infeccioses i Resposta Inflamatòria Sistèmica en Pediatria, Unitat d’Infeccions, Servei de Pediatria, Institut de Recerca Pediàtrica Sant Joan de Déu, 08950 Barcelona, Spain; 4Molecular Genetics Department, Hospital Sant Joan de Déu, 08950 Barcelona, Spain; 5Pediatric Infectious Diseases, Rheumatology and Immunology Unit, Hospital Universitario Virgen del Rocío, Instituto de Biomedicina de Sevilla, 41013 Seville, Spain; 6Red de Investigación Traslacional en Infectología Pediátrica RITIP, 28029 Madrid, Spain; 7Department of Infectious Diseases, Hospital 12 de Octubre, 28041 Madrid, Spain; 8Pediatrics Department, Hospital de Getafe, 28905 Madrid, Spain; 9Centro de Investigación Biomédica en Red de Enfermedades Infecciosas (CIBERINFEC), 28029 Madrid, Spain; 10Pediatrics and Infectious Diseases Unit, La Paz University Hospital, 28046 Madrid, Spain; 11Neumología Pediátrica, Servicio de Pediatría, Hospital Universitario Jerez de la Frontera, 11407 Cádiz, Spain; 12Pediatrics Department, Fundación Jiménez Díaz, 28040 Madrid, Spain; 13Pediatrics Department, Hospital Joan XXIII, 43005 Tarragona, Spain; 14Servicio de Pediatría, Hospital Clinico San Carlos e Instituto de Investigación Sanitaria del Hospital Clínico San Carlos, 28040 Madrid, Spain; 15Departamento de Salud Pública y Materno-Infantil, Universidad Complutense de Madrid, 28040 Madrid, Spain; 16Pediatric Infectious Diseases and Immunodeficiencies Unit–Drassanes Unit, Hospital Vall D’Hebron, 08035 Barcelona, Spain; 17Sección Enfermedades Infecciosas Pediátricas, Servicio de Pediatría, Hospital Gregorio Marañón, 28009 Madrid, Spain; 18Instituto de Investigación Sanitaria Gregorio Marañon, 28009 Madrid, Spain; 19Centro de Investigación Biomédica en Red de Epidemiología y Salud Pública (CIBERESP), 28029 Madrid, Spain; 20Departament de Cirurgia i Especialitats Medicoquirúrgiques, Facultat de Medicina i Ciències de la Salut, Universitat de Barcelona, 08036 Barcelona, Spain; 21Pharmacy Department, Division of Medicines, Hospital Clinic of Barcelona, University of Barcelona, 08036 Barcelona, Spain; 22Passeig Sant Joan de Déu 2, 08950 Esplugues, Spain

**Keywords:** acetylation, infant, isoniazid, pharmacokinetics, transaminases, tuberculosis

## Abstract

In 2010, the WHO recommended an increase in the daily doses of first-line anti-tuberculosis medicines in children. We aim to characterize the pharmacokinetics of the once-daily isoniazid (INH) dose at 10 mg/kg of body weight in infants <6 months of age. We performed a multicenter pharmacokinetic study in Spain. The N-acetyltransferase 2 gene was analyzed to determine the acetylation status. Samples were analyzed using a validated UPLC-UV assay. A non-compartmental pharmacokinetic analysis was performed. Twenty-three pharmacokinetic profiles were performed in 20 infants (8 females) at a median (IQR) age of 19.0 (12.6–23.3) weeks. The acetylator statuses were homozygous fast (*n* = 1), heterozygous intermediate (*n* = 12), and homozygous slow (*n* = 7). INH median (IQR) C_max_ and AUC_0–24h_ values were 4.8 (3.7–6.7) mg/L and 23.5 (13.4–36.7) h*mg/L and the adult targets (>3 mg/L and 11.6–26.3 h*mg/L) were not reached in three and five cases, respectively. The age at assessment or acetylator status had no impact on C_max_ values, but a larger INH AUC_0–24h_ (*p* = 0.025) and trends towards a longer half-life (*p* = 0.055) and slower clearance (*p* = 0.070) were observed in homozygous slow acetylators. Treatment was well tolerated; mildly elevated alanine aminotransferase levels were observed in three cases. In our series of young infants receiving isoniazid, no major safety concerns were raised, and the target adult levels were reached in most patients.

## 1. Introduction

In 2021, it was estimated that 10.6 million people fell ill with tuberculosis (TB) and 1.6 million died [1]. TB was the leading cause of death from a single infectious agent until 2019 worldwide, but COVID-19 has taken over the lead recently [1]. Children (aged < 15 years) account for 12% of new TB cases, half of which are estimated to occur in children aged <5 years [2]. In Spain, the incidence of TB has been decreasing over the last 20 years and was 9.3 cases per 100,000 persons in the general population, and 4.2 per 100,000 in children <15 years-of-age in 2019 [3]. In infants and toddlers, the risk of progression from primary infection to disease inversely correlates with age. TB usually develops within the first year after primary infection, and severe and disseminated forms are more frequent than in older children or adults [4]. Young children represent 80% of TB deaths among patients <15 years of age. Up to two thirds of paediatric TB cases are underdiagnosed and underreported [2].

Pharmacokinetic/pharmacodynamic data assessing the association between drug concentrations and clinical outcomes in children with TB are lacking. Therefore, maximum plasma or serum drug concentrations (C_max_) from adult studies have been used as surrogate markers of optimal anti-TB therapy dosages in children. The target C_max_ for isoniazid (INH) ranges from 3 to 6 µg/mL [5]. The total exposure to INH (area under the concentration versus time curve, AUC_0–24h_) is probably more relevant to the efficacy of INH and other first-line TB drugs, based on studies in mice and in the hollow fiber model [6,7]. In September 2009, the recommended dosages of first-line oral anti-TB drugs for children were updated by the World Health Organization (WHO) following a systematic review of the literature that demonstrated that children require higher doses than adults to reach the same serum drug concentrations [8]. The isoniazid (INH) daily dosage was increased from 5 (range: 4–6) to 10 (10–15) mg/kg of body weight, both for the treatment of TB and for preventive therapy, with a maximum daily dosage of 300 mg [8]. The INH dosing range was later broadened to 7–15 mg/kg [9]. 

The approach to drug dosing in infants and children is based on the physiological characteristics of the patient, which continuously evolve during the first years of life, and the pharmacokinetic parameters of the drug [10]. In the case of INH, serum levels are also influenced by genetic polymorphisms of N-acetyltransferase type 2 (*NAT2*) [11]. In young infants with TB, standard drug dosages and treatment regimens are recommended, but the evidence for this recommendation is of low quality [9]. Several studies have investigated the pharmacokinetics of anti-TB drugs in neonates and young infants, exclusively in high TB burden countries [12,13,14,15,16,17]. 

In this study, we aimed to determine the pharmacokinetics of INH and the effect of age and *NAT2* acetylator status in a cohort of young infants in Spain, a low TB burden country. 

## 2. Results

Twenty-three pharmacokinetic profiles were performed in 20 infants (8 females, 40%) treated for TB exposure with INH preventive therapy (*n* = 14) or TB disease (*n* = 6) (Table 1). Only two of them (10%) were premature and had a birth weight <1.5 kg; both received INH preventive therapy. None of the patients were HIV-exposed. No other comorbidities were observed. Baseline ALT, hemoglobin, and albumin levels were within normal limits in all cases. Three infants on INH preventive therapy were sampled twice, before and after 3 months of age. In two patients, the blood sample at the 3rd time-point could not be obtained and, therefore, t1/2 and Cl/F could not be determined. 

On the day of the pharmacokinetic assessment, the median (IQR) age was 19.0 (12.6–23.3) weeks, and the median (IQR) weight for the age was 42.7% (16.5–68.5); the weight for the age was below percentile 5 in four patients, including the two premature babies and two infants treated for TB disease. Besides INH, the six patients affected with TB disease were receiving other anti-TB drugs and three of them were also being treated with steroids. According to the *NAT2* genotype, the acetylator statuses were FF (*n* = 1), FS (*n* = 12), and SS (*n* = 7). Adherence to anti-TB drugs in the 3 previous days was 100% and tolerance to treatment was optimal in all cases. Median (IQR) pre-dose and post-dose fasting times were 230 (189–520) and 40 (28–74) minutes, respectively. 

The estimated pharmacokinetic parameters are summarized in Table 2. Wide inter-individual variability in pharmacokinetics was observed. INH C_max_ ranged from 0.98 to 13.8 mg/L and the >3 mg/L target recommended in adults was not reached in three cases (13.0%); no significant differences in any of the variables were observed when comparing those patients that reached the 3 mg/L target and those that did not (data not shown). The median (IQR) AUC_0–24h_ of INH was 23.5 (13.4–36.7) h*mg/L, and the lower AUC_0–24h_ of the reference values in adults (11.6–26.3 h*mg/L) was reached in 18 cases (78.3%). 

Gender, age at assessment sampling (<3 or >3 months), treatment with anti-TB drugs other than INH or breastfeeding had no impact on pharmacokinetic parameters (Table 3). A larger INH AUC_0–24h_ (*p* = 0.025) was observed among the SS *NAT2* genotype patients, together with trends towards longer t1/2 (*p* = 0.055) and slower Cl/F (*p* = 0.070) (Figure 1). 

Significant correlations were observed between the weight for age and C_max_ (*r* = 0.504, *p* = 0.014; Spearman’s rank test) and T_max_ (*r* = −0.431, *p* = 0.040); and length for age and C_max_ (*r* = 0.452, *p* = 0.030), T_max_ (*r* = −0.429, *p* = 0.041), and AUC_0–24h_ (*r* = 0.423, *p* = 0.044); the weight for length did not show any significant correlations. Longer pre-dose fasting times correlated with higher AUC_0–24h_ values (*r* = 0.530, *p* = 0.009), and a trend was also observed for C_max_ values (*r* = 0.410, *p* = 0.052). The post-dose fasting time had no impact on the pharmacokinetic parameters. 

On the sampling day, mildly elevated non-symptomatic levels of ALT (range: 61–76 UI/L; <2.5 times above the upper limit of normal, grade 1 toxicity [18]) were observed in three infants (out of 22 curves, 13.6%), two of whom were treated for TB disease and one of them was on preventive therapy. The median C_max_ and AUC_0–24h_ values of these three infants (3.2 mg/L and 14.3 h*mg/L) were not significantly different from those of the other patients (4.8 mg/L and 27.7 h*mg/L; *p* = 0.408 and *p* = 0.265, respectively). ALT normalized in all cases after treatment interruption. No other toxicities were observed. 

## 3. Discussion

To our knowledge, this is the first study describing the pharmacokinetics of the revised WHO-recommended doses of INH in young infants in a low TB burden country. As compared with previous studies in high TB burden regions, the cohort we report is different in terms of age and ethnic background, but also with regards to other clinical covariates that may influence the pharmacokinetics of the drug, most importantly the indication for INH treatment, HIV coinfection rates, and nutritional status [19]. In our study, most infants were well-nourished at the time of assessment, none of them were HIV-infected or HIV-exposed, and preventive therapy with INH monotherapy after contact with a smear-positive TB index case was the most common indication for INH treatment. Our study nicely reflects the nature of the pediatric TB epidemic in low burden countries, such as Spain, where most children are healthy and identified in contact tracing studies in early stages of infection [20]. 

Several studies in high TB burden countries have analyzed the pharmacokinetics of the 10 mg/kg WHO-recommended dose of INH in older infants and toddlers, and serum INH C_max_ above the 3 mg/L threshold have been reported in most cases to date (range: 83–100% [13,21,22,23,24,25]), matching the target adult values [5]. Four studies in neonates and young infants have also validated the new recommendation for INH dosage in this age range. In a cohort of 20 low birth weight and premature neonates (median daily INH dose: 10 mg/kg), only one of them did not reach the 3 mg/L target [12]. In another study from the same South-African group, all 39 infants aged below 12 months with TB (median daily INH dose: 12.8 mg/kg) achieved the target adult peak of INH [13]. Similarly, a study in Malawi and South-Africa (median daily INH dose: 12.0 mg/kg) observed INH exposure within the AUC_0–24h_ target range in older infants, and above the target for children below 3 months of age [16]. Additionally, a large clinical trial in South-African HIV-exposed or HIV-infected children reported an INH C_max_ > 3 mg/L in 99% of 151 infants aged 3 months (mean daily INH dose: 14.5 mg/kg) [17]. As opposed to the former, a pharmacokinetic substudy of the SHINE trial, the first to use the dispersible child-friendly fixed-dose combinations of anti-TB drugs (MacLeods Pharmaceuticals; India), recently reported lower than expected INH C_max_ and AUC_0–24h_ values in infants (*n* = 16, median age: 0.6 years) in the lowest weight band (4.0 to 7.9 kg). Although comprehensive data of these patients are not given, most of the former weighed near the upper end of the weight band and were actually underdosed (median daily INH dose: 7.1 mg/kg) [15]. 

In our cohort of young infants, the target INH C_max_ was also observed in most cases (87%), with drug levels in the low range of those previously reported [14,21,22]. With regards to AUC_0–24h_, most children reached enough total exposure to INH based on adult pharmacokinetic data as well [26]. We could not identify any risk factor for failure to reach the INH C_max_ or AUC_0–24h_ target values. Very similar to previous studies, gender, the indication for INH treatment, the concomitant use of other anti-TB drugs, including rifampicin in all cases, and the feeding type had no impact on the pharmacokinetic parameters that were assessed in our cohort [13,14,16]. The pre-dose fasting time did however correlate with C_max_ and AUC_0–24h_ values in our study. A fasting state is recommended to increase INH bioavailability, but this is not always feasible in the youngest infants, in whom drug absorption may therefore be compromised [27,28]. These findings further emphasize the need to administer INH on an empty stomach. 

The elimination of INH in young children is both influenced by age and the maturation of the trimodal *NAT2* acetylation pathway within the first years of life, which impacts INH metabolism from birth, despite immaturity [11,29]. As previously reported in neonates [12], we observed higher INH exposure in infants with SS *NAT2* genotypes as compared to FS genotypes, and in FS genotypes as compared to FF genotypes, although these findings were only statistically significant for AUC_0–24h_ levels (median values: 35.7, 21.4 and 6.1 h*mg/L, respectively), probably because of low numbers. In our cohort, age (<3 or >3 months) did not influence INH pharmacokinetics, although relevant differences may have gone unnoticed given the stringent upper 6-month age limit that we used, resulting in a small age range. Higher drug exposures are expected in infants aged below 3 months due to incomplete enzyme maturation, as demonstrated by Denti et al. in their simulations [16]. 

Data describing the potential toxicity of the newly WHO-recommended dosages of first-line anti-TB drugs in neonates and young infants are still scarce, especially with regards to hepatic toxicity associated with regimens combining different anti-TB drugs. It should be noted that INH is prescribed together with rifampicin and pyrazinamide in the 2-month intensive phase of the treatment of drug-sensitive TB, and the three drugs known to be hepatotoxic [27]. While further data on toxicity are needed, our results and those of others are reassuring, with the self-limited symptom-free elevation of ALT being the only adverse event reported to date in HIV-uninfected infants [12,13]. 

Our study is limited by low numbers, which may be too low to reveal predictors for low exposure. Conducting pharmacokinetic studies in infants is challenging because of low study consent rates, difficulty in obtaining blood, and the limited blood volume available. Additionally, the new child-friendly fixed-dose combinations are not available yet in Europe and we had to use crushed INH tablets, as families with children with TB are forced to do routinely in Spain [30,31]. Furthermore, drug intake in very young infants is challenging due to their immature ability to swallow medications and palatability issues [32]. In previous studies in neonates and young infants, for instance, INH was given through a nasogastric tube [12,13]. We did our best to optimize INH administration on the day of pharmacokinetic assessment, but some loss of medication from spillage or spitting cannot be completely excluded. Finally, we only obtained three samples from each child, which is a limitation when it comes to determining C_max_ and estimating AUC_0–24h_. 

In summary, our results are consistent with those of previous studies in high TB burden countries and endorse the 10 mg/kg of daily INH recommendation in infants below the age of 6 months, which is safe and ensures appropriate INH serum concentrations in most children, irrespective of the acetylator status. Further studies assessing the pharmacokinetics and safety of the child-friendly formulations of anti-TB drugs at WHO-recommended doses in young infants are needed, with a special focus on the most vulnerable patients, such as those with disseminated TB or HIV co-infection. 

## 4. Materials and Methods

### 4.1. Design and Setting

We conducted a national multicenter pharmacokinetic study within centers participating in the Spanish Pediatric TB Research Network [33,34]. The ethics approval was obtained from the Hospital Sant Joan de Déu (Barcelona, Spain) Ethics Committee (reference EPA-05-014) and from every participating center ethics committee thereafter. Written informed consent for participation was obtained from the parents/legal guardians of each participant at inclusion.

### 4.2. Study Procedures

Infants aged below 6 months of age were consecutively recruited if receiving daily oral INH (10 mg/kg/day) for at least the previous 7 days, either in preventive therapy after contact with a smear-positive TB index case or TB infection, or in TB disease treatment with other anti-TB drugs, as per national guidelines [27]. In Spain, patients without clinical and radiological signs or symptoms consistent with TB and a positive immunodiagnostic TB test are diagnosed with TB infection, while the diagnosis of TB disease is based on epidemiological, clinical, radiological, and microbiological findings, according to published consensus criteria [35]. Infants were not eligible in the case of a postconceptional age below 40 weeks on the day of the pharmacokinetic assessment, baseline (before INH implementation) alanine aminotransferase levels (ALT) >50 IU/L, or when affected with any renal or hepatic disease. Infants were allowed to participate twice in the study (before and after 3 months of age) if they fulfilled inclusion criteria; in this case, each sampling curve was analyzed individually. 

The following data were collected from all participants: gender, ethnicity, weight, and gestational age at birth, the indication for INH treatment, and whether or not the patient was receiving other drugs, the feeding type, and, in case of breastfeeding, whether the mother was also receiving INH or not, and the baseline ALT, hemoglobin, and albumin levels; on the day of the pharmacokinetic sampling, data including the age, weight, and length for age (measured naked by the investigator), and weight for length using the “WHO Child Growth Standards” charts (http://www.who.int/childgrowth/standards/en/ accessed on 18 October 2022), and weeks on INH treatment were collected. Adherence to anti-TB treatment was reinforced through a nurse-led educational intervention and was assessed by means of a written questionnaire that the parents had to complete on each of the 3 days before the pharmacokinetic assessment [36].

### 4.3. Drug Administration, Pharmacokinetic Sampling and Laboratory Analysis

On the day of the pharmacokinetic study, the INH dose was prepared by crushing Cemidon^®^ 50 B6 tablets (Chiesi España; Barcelona, Spain); each tablet contains 50 mg of INH and 15 mg of pyridoxine. Cemidon^®^ tablets are currently the only INH formulation licensed for all pediatric groups (other than fixed-dose combinations not licensed for infants) for the treatment of TB in Spain [30]. The INH dose (10 mg/kg) was accurately weighted from the crushed tablets and rounded up to the nearest mg (i.e., an infant weighing 6.66 kg would receive a 67 mg INH dose). The dose was dissolved in 1 mL of water prior to oral administration under the supervision of the investigators. Pre-dose (at least for 3 h) and post-dose (at least for 20 min) fasting periods were recommended.

The INH dose, the time of INH administration, and the fasting periods before and after INH administration were recorded. Three venous blood specimens of 0.5 mL each were obtained at 1, 3, and 6 h post-dose or at 2, 4, and 8 h post-dose; infants were alternatively assigned to one of the schedules, according to the order of inclusion. Blood samples were collected in serum separator tubes and kept on ice until centrifugation within 30 min of collection. Next, 200 µL sera aliquots were frozen at −80 °C and transported to the Netherlands on dry ice. INH was measured with liquid-liquid extraction, followed by ultra-performance liquid chromatography with UV detection. The accuracy was between 97.8% and 106.7%, depending on the concentration level. The within-day and between-day coefficients of variation were less than 13.4% and less than 3.2% (depending on the concentration), respectively, over the range of 0.05 to 15.1 mg/liter. Lower limits of quantification for INH were 0.05 mg/liter.

The remaining blood cells were used to extract DNA using standard procedures for *NAT2* genotyping [37]. The DNA samples were screened for the different polymorphisms previously described in the *NAT2* gene. Primer design was performed, according to the *NAT2* gene RefSeq accession number NM_000015.2. All the 873 coding nucleotides were screened directly by PCR amplification and sequenced using capillary electrophoresis on an ABI Prism 3130 Genetic Analyzer (Applied Biosystems Inc.; Foster City, CA, USA). The results were analyzed using sequencing analysis software version 6 (Applied Biosystems Inc.; Foster City, CA, USA). The numbering of nucleotides was based on the cDNA RefSeq sequence, with the A of the ATG start codon marked as +1. According to Vatsis nomenclature [38], the wild-type fast allele (F) was assigned as *NAT2*4, NAT2*12,* or *NAT2*13*. These alleles confer normal enzyme activity on the *NAT2* protein, while the mutant slow alleles (S), classified as *NAT2*5, NAT2*6, NAT2*7,* and *NAT2*14* in humans, confer decreased enzyme activity on the *NAT2* protein. Depending on the allele combination observed, the study participants were classified as homozygous fast (FF), heterozygous intermediate (FS), or homozygous slow (SS) acetylators.

### 4.4. Pharmacokinetic Parameters and Statistical Analysis

Pharmacokinetic parameters were estimated using non-compartmental analysis (WinNonLin version 5.3, Pharsight Corp.; Mountain View, CA, USA). The highest observed plasma concentration was defined as C_max_ with the corresponding sampling time as T_max_. Due to the limited number of sampling points, the log-linear period (log C versus t) was based on two data points (3 and 6 h or 4 and 8 h). The absolute value of the slope (β/2.303, in which β is the first order elimination rate constant) was calculated using linear regression analysis. The elimination half-life (*t*_½_) was calculated as 0.693/β. Concentrations beyond 6 or 8 h and up to 24 h were estimated based on the decay of concentrations based on first-order pharmacokinetics described by the formula C_T2_ = C_T1_ × e^−β×(T2−T1)^. The area under the plasma concentration-time curve (AUC_0–24h_) was calculated using the linear/log-trapezoidal rule from zero up to the last concentration at 24 h. The apparent clearance of the drug (Cl/F; in which F is bioavailability) was calculated as dose/AUC_0–24h_ and the apparent volume of distribution (Vd/F) was calculated as (Cl/F)/β. The reference range for INH C_max_ was 3 to 6 mg/L [5]. The range in median INH AUC_0–24h_ values in the studies evaluated in a systematic review (11.6–26.3 h*mg/L, excluding one study with outlying results) was used as a reference for total exposure to INH [26], as was carried out by Chabala et al. [15]. Pharmacokinetic parameters were summarized using medians and interquartile ranges (IQR) and were compared categorically by gender, the patient´s age at the time of pharmacokinetic sampling (<3 or >3 months), *NAT2* genotype, concomitant use of other anti-TB drugs, and breastfeeding, using the Mann–Whitney U test or the Kruskal–Wallis test. Correlations between numerical variables were calculated using the non-parametric Spearman´s rank correlation coefficient. Statistical analysis was carried out using SPSS version 21.0 (IBM Corp.; Armonk, NY). The statistical significance was defined as a *p*-value < 0.05.

## Figures and Tables

**Figure 1 antibiotics-12-00272-f001:**
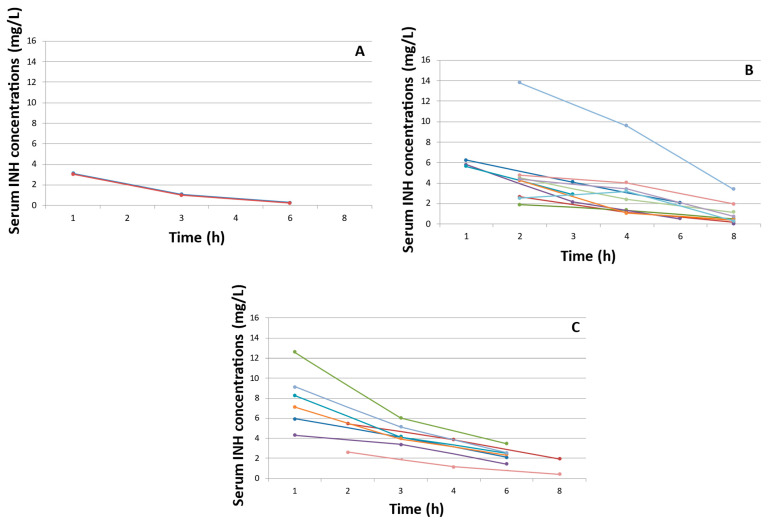
Isoniazid serum concentrations (in mg/L) according to *NAT2* genotyping by acetylator type. (**A**) *NAT2* homozygous fast genotype; (**B**) *NAT2* heterozygous intermediate genotype; and (**C**) *NAT2* homozygous slow genotype.

**Table 1 antibiotics-12-00272-t001:** Clinical characteristics of the infants (*n* = 20) included to assess isoniazid pharmacokinetics at sampling (*n* = 23); data are shown either as n (%) or as median (interquartile range).

Infants’ Characteristics (*n* = 20)	
Sex (female)	8 (40)
Mother and/or father of Caucasian ethnicity ^a^	10 (50)
Birth weight (kg)	3.1 (2.7–3.5)
Gestational age (weeks)	39.1 (38.6–39.9)
Indication for INH treatment	
Preventive therapy	14 (70)
TB disease ^b^	6 (30)
Before INH treatment implementation	
ALT levels (IU/L)	22 (15–28)
Hemoglobin levels (g/dL)	11.0 (10.3–11.9)
Abnormal hemoglobin levels [18]	0
Albumin levels (g/L)	40.3 (39.1–43.1)
Abnormal albumin levels [18]	0
On the day of pharmacokinetic sampling (*n* = 23)	
Age (weeks)	19.0 (12.6–23.3)
<3 months of age	8 (34.8)
>3 months of age	15 (65.2)
Weight (kg)	6.5 (5.3–7.6)
Weight for age (percentile)	42.7 (16.5–68.5)
Length (cm)	62.3 (56.6–65.8)
Length for age (percentile)	28.6 (11.3–61.0)
Weight for length (percentile)	44.1 (21.7–84.1)
Breastfeeding ^c^	7 (30.4)
INH dose (mg/kg)	10.1 (10.0–10.1)
Concomitant medications	
Other anti-TB drugs ^d^	6 (26.1)
Steroids	3 (13.0)
Time on INH treatment (weeks)	4.1 (2.9–8.5)
ALT levels (IU/L)	29.5 (19.3–37.5)
ALT levels > 50 IU/L	3 (13.0)
Pre-dose fasting time (minutes)	230 (189–520)
Post-dose fasting time (minutes)	40 (28–74)

ALT, alanine aminotransferase; INH, isoniazid; TB, tuberculosis. ^a^ Caucasian parents (*n* = 7), Latin American parents (*n* = 5), Caucasian father and Latin American mother (*n* = 3), Black parents (*n* = 2), Gypsy’s parents (*n* = 2), and Asian parents (*n* = 1). ^b^ Including pulmonary TB (*n* = 3), pulmonary TB with central nervous system involvement (*n* = 2), and miliary TB (*n* = 1). ^c^ Only one out of 7 breastfeeding mothers was also receiving anti-TB treatment for herself (INH and rifampicin). ^d^ On rifampicin, pyrazinamide, and ethambutol (*n* = 4) and rifampicin, pyrazinamide, and amikacin (*n* = 1) in the intensive phase of TB treatment; and rifampicin (*n* = 1) in the continuation phase of TB treatment.

**Table 2 antibiotics-12-00272-t002:** Summary of estimated pharmacokinetic parameters of 23 assessments in 20 infants; 3 patients on preventive therapy were sampled twice, before and after 3 months of age. All data shown as median (interquartile range).

	Median (IQR)
C_max_ (mg/L)	4.8 (3.7–6.7)
T_max_ (hours)	1.2 (1.0–2.0)
AUC_0–24h_ (h*mg/L)	23.5 (13.4–36.7)
*t*_1/2_ (hours)	2.9 (2.2–3.2)
Cl/F (L/hours)	2.4 (1.8–4.2)
Vd/F (L)	10.0 (7.7–13.1)

AUC_0–24h_, area under the concentration-time curve during the dosing interval; Cl/F, apparent clearance of the drug; C_max_, maximum drug concentration; IQR, interquartile range; *t*_1/2_, half-life; T_max_, time to C_max_; Vd/F, apparent volume of distribution.

**Table 3 antibiotics-12-00272-t003:** Effect of different clinical characteristics on serum isoniazid pharmacokinetic parameters in the 23 assessments performed; all data are shown as medians (interquartile range). All data were analyzed using the Mann–Whitney U test, except for *NAT2* genotype (Kruskal–Wallis test).

	N	C_max_ (mg/L)	*p*	T_max_ (h)	*p*	AUC_0–24h_ (h*mg/L)	*p*	n	*t*_1/2_ (h)	*p*	Cl/F (L/h)	*p*	Vd/F (L)	*p*
Female	9	5.8 (4.8–7.18)	0.439	1.2 (1.0–2.0)	0.516	27.7 (14.5–34.7)	0.643	8	3.1 (2.7–3.6)	0.268	1.9 (1.6–3.0)	0.268	9.2 (7.5–11.0)	0.238
Male	14	4.4 (3.4–6.1)		1.2 (1.0–2.0)		22.2 (11.4–37.6)		13	2.7 (1.8–3.1)		3.3 (1.8–4.2)		10.6 (9.9–14.5)	
<3 months	7	5.5 (4.0–6.1)	0.922	1.1 (1.0–1.6)	0.341	27.7 (18.3–33.4)	0.922	7	3.1 (1.9–3.6)	0.913	1.9 (1.7–3.4)	0.585	8.7 (7.7–10.2)	0.255
>3 months	16	4.6 (4.0–7.4)		1.6 (1.0–2.0)		22.2 (13.8–36.9)		14	2.9 (2.3–3.2)		3.0 (1.9–4.1)		11.2 (9.8–14.1)	
*NAT2* genotype FF	2	3.1 (3.0–3.1)	0.136	1.0 (1.0–1.0)	0.282	6.1 (5.3–6.9)	0.025 *	2	1.4 (1.3–1.4)	0.055	12.2 (9.6–14.8)	0.070	23.5 (19.0–28.0)	0.198
*NAT2* genotype FS	13	4.8 (4.2–5.8)		2.0 (1.1–2.0)		21.4 (14.3–27.7)		11	2.9 (2.0–3.3)		3.3 (1.6–4.0)		9.9 (7.7–12.5)	
*NAT2* genotype SS	8	6.5 (5.2–8.5)		1.2 (1.1–1.4)		35.7 (28.4–38.2)		8	3.1 (2.9–3.4)		2.1 (1.8–2.5)		10.3 (8.3–11.3)	
INH monotherapy	17	5.6 (4.4–7.1)	0.177	1.1 (1.0–2.0)	0.201	27.7 (14.5–36.7)	0.431	16	2.8 (2.1–3.2)	0.603	2.1 (1.8–5.7)	0.660	10.0 (7.8–12.0)	0.313
Combined therapy	6	3.8 (2.3–4.5)		2.0 (1.4–3.4)		18.6 (10.6–23.4)		5	3.1 (2.9–3.1)		3.3 (2.8–4.8)		12.8 (9.9–14.3)	
Breastfeeding	7	4.8 (4.5–6.0)	0.871	2.0 (1.1–2.0)	0.492	27.2 (14.4–32.1)	0.974	6	2.8 (1.9–3.7)	0.910	1.8 (1.5–3.3)	0.340	9.8 (7.8–12.0)	0.095
Not breastfeeding	16	5.0 (2.8–7.4)		1.2 (1.0–2.0)		23.2 (10.7–37.0)		15	2.9 (2.5–3.2)		2.8 (1.9–4.5)		10.6 (8.7–14.5)	

(*) Post hoc Mann–Whitney U test results: SS versus FS, *p* = 0.121; SS versus FF, *p* = 0.044; and FS vs. FF, *p* = 0.019. AUC_0–24h_, area under the concentration-time curve during the dosing interval; Cl/F, clearance of the drug; C_max_, maximum drug concentration; FF, homozygous fast; FS, heterozygous intermediate; INH, isoniazid; IQR, interquartile range; *NAT2*, N-acetyltransferase 2; SS, homozygous slow; *t*_1/2_, half-life; T_max_, time to C_max_; Vd/F, apparent volume of distribution.

## Data Availability

The data underlying this article will be shared on reasonable request to the corresponding author.

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
