# Peer review of "Adequacy of the 10 mg/kg Daily Dose of Antituberculosis Drug Isoniazid in Infants under 6 Months of Age"

_antibiotics, 2023, doi:10.3390/antibiotics12020272_

Round 1

Reviewer 1 Report

In this study, the authors investigated the adequacy of the 10 mg/kg daily dose of isoniazid in infants under 6 months of age. Although the study represents interesting findings regarding he nature of the pediatric TB epidemic in low burden countries such as Spain. However, many points were not clearly addressed in their work as follows:

1)     The major limitation of this study is the limited number of cases investigated.

2) More explanation regarding the effect of pre-dose fasting time on Cmax and AUC0-24h values should be included.

3) Why acetylator status had no impact on Cmax values but exerted significant effect on AUC0-24h. The authors should clearly explain such discrepancy

4) How the authors ensured the adherence of all studied cases to the proposed dose of INH (10 mg/kg), given that INH was administered in the form of crushed tablets

Author Response

In this study, the authors investigated the adequacy of the 10 mg/kg daily dose of isoniazid in infants under 6 months of age. Although the study represents interesting findings regarding he nature of the pediatric TB epidemic in low burden countries such as Spain.

We thank the reviewer for the thorough revision of our manuscript. We did our best to address the suggestions and agree that the paper now reads better.

However, many points were not clearly addressed in their work as follows: 

  • The major limitation of this study is the limited number of cases investigated. We totally agree with the reviewer and this has been acknowledged in the Discussion (limitations para). Previous similar studies have also investigated low number of patients. We have added the following sentence in the text to further emphasize this (line 247):

Conducting pharmacokinetic studies in infants is challenging because of low study con-sent rates, difficulty in obtaining blood and limited blood volume available.

  • More explanation regarding the effect of pre-dose fasting time on Cmax and AUC0-24h values should be included. The bioavailability of orally administered isoniazid is increased in a fasting state and this is universally recommended as standard of care. In our study in young infants (line 304), and also in clinical practice, we routinely recommend a previous fasting period of 3 hours minimum before isoniazid is given.

Actually, median (IQR) pre-dose fasting time was 230 (189-520) minutes in our study. The fact that both Cmax and AUC0-24h significantly correlated with pre-dose fasting time strengths this recommendation. Therefore, we have added the following sentence in the Discussion (line 224):

These findings further emphasize the need to administer INH on an empty stomach. 

  • Why acetylator status had no impact on Cmax values but exerted significant effect on AUC0-24h. The authors should clearly explain such discrepancy. These findings are most likely related to low numbers. As previously commented, it is often very difficult to enroll young infants in PK studies. Actually, although not statistically significant, Cmax progressively increased according to NAT 2 genotypes: 3.1, 4.8 and 6.5 mg/L in FF, FS and SS genotypes, respectively. The sentence referring to these findings in the Discussion has been rephrased (starting in line 228):

As previously reported in neonates [11], we observed higher INH exposure in infants with SS NAT2 genotypes as compared to FS genotypes, and in FS genotypes as compared to FF genotypes, although these findings were only statistically significant for AUC0-24h levels (median values: 35.7, 21.4 and 6.1 h*mg/L, respectively), probably because of low numbers.  

How the authors ensured the adherence of all studied cases to the proposed dose of INH (10 mg/kg), given that INH was administered in the form of crushed tablets. Thanks for pointing this out. Adherence to anti-TB treatment is critical for disease cure, and the lack of child-friendly formulations and the need to crush available tablets were additional challenges to optimize adherence in our study. In this case, we used a nurse-led educational intervention (consisting of leaflets with written information about the disease and the importance of adherence in the child’s native language and follow-up telephone calls) and we assessed adherence by means of a written questionaire. The results of this strategy have previously been published:

Guix-Comellas EM, Rozas-Quesada L, Velasco-Arnaiz E, Ferrés-Canals A, Estrada-Masllorens JM, Force-Sanmartín E, Noguera-Julian A. Impact of nursing interventions on adherence to treatment with antituberculosis drugs in children and young people: A nonrandomized controlled trial. J Adv Nurs. 2018 May 3. doi: 10.1111/jan.13692. Epub ahead of print. PMID: 29726024.

Guix-Comellas EM, Rozas L, Velasco-Arnaiz E, Morín-Fraile V, Force-Sanmartín E, Noguera-Julian A. Adherence to Antituberculosis Drugs in Children and Adolescents in A Low-Endemic Setting: A Retrospective Series. Pediatr Infect Dis J. 2017 Jun;36(6):616-618. doi: 10.1097/INF.0000000000001508. PMID: 28030525.

Guix-Comellas EM, Rozas-Quesada L, Force-Sanmartín E, Estrada-Masllorens JM, Galimany-Masclans J, Noguera-Julian A. Influence of nursing interventions on adherence to treatment with antituberculosis drugs in children and young people: research protocol. J Adv Nurs. 2015 Sep;71(9):2189-99. doi: 10.1111/jan.12656. Epub 2015 Mar 26. PMID: 25818512.

To further explain this strategy, the following sentence in the Methods has been rephrased (starting in line 293):

Adherence to anti-TB treatment was reinforced through a nurse-led educational interven-tion and was assessed by means of a written questionnaire that the parents had to com-plete at each of the 3 days before the pharmacokinetic assessment [36].

Reviewer 2 Report

Dear Editor,

Many thanks for inviting me to review this paper. This study investigated “Adequacy of the 10 mg/kg daily dose of antituberculosis drug 2 isoniazid in infants under 6 months of age” I write my suggestions below.

I believe this study aligns with the scope of the journal. Antibiotics is a highly reputable academic journal and has a distinguished audience. And its’ audience deserve high-quality and exquisite publications.

I believe this study fills an important in the clinical practice. I believe this study is qualified enough to published as it is at Journal of Antibiotics.

·                I believe the title is suitable and adheres with the content of the study.

·                I believe the introduction section is well organized and beneficial.

·                The methods and study framework are quite solid. To increase the impact of the paper and readability of the text could be improved.

·                The methods section is rather long. I believe author might shorten up the methods.

·                The take home message can be underlined at the discussion section.

·                Give some details about the outcomes could increase the readability and scientific soundness.

·                Please adhere the journal guideline especially for reference sections.

·                It would be better to give up to date references.

·                Keywords: I would like to recommend the adhere MeSH headings. 

Author Response

Many thanks for inviting me to review this paper. This study investigated “Adequacy of the 10 mg/kg daily dose of antituberculosis drug 2 isoniazid in infants under 6 months of age” I write my suggestions below. 

I believe this study aligns with the scope of the journal. Antibiotics is a highly reputable academic journal and has a distinguished audience. And its’ audience deserve high-quality and exquisite publications. 

I believe this study fills an important in the clinical practice. I believe this study is qualified enough to published as it is at Journal of Antibiotics. 

We thank the reviewer for the very positive comments and interesting suggestions on our manuscript. We tried to address the latter when possible and we strongly believe the paper now reads better.

  • I believe the title is suitable and adheres with the content of the study.
  • I believe the introduction section is well organized and beneficial.
  • The methods and study framework are quite solid. To increase the impact of the paper and readability of the text could be improved.The methods section is rather long. I believe author might shorten up the methods. We acknowledge the Methods section is long. At the same time, we strongly believe it is vital to give comprehensive details on how the study was designed and performed, so that other coalleagues may replicate it. Actually, one of the reasons to choose this journal was that it has no word limit. 
  • The take home message can be underlined at the discussion section.Thanks for the suggestion. The Conclusions para, in the Discussion, underlines the take home message, as follows (line 258):

In summary, our results are consistent with those of previous studies in high TB burden countries and endorse the 10 mg/kg of daily INH recommendation in infants be-low the age of 6 months, which is safe and ensures appropriate INH serum concentrations in most children, irrespective of the acetylator status.

  • Give some details about the outcomes could increase the readability and scientific soundness.This was a multicenter cross-sectional study while the patients were receiving isoniazid. Most of the patients were treated as per preventive treatment after contact with a smear-positive adult, but were not ill. Unfortunately, we did not collect data on follow-up of the patients with tuberculosis disease that took part in the study and cannot answer this query.
  • Please adhere the journal guideline especially for reference sections.Thanks. We have doublie-checked that the references fulfilled the requirements of the journal and spotted several typos, that have been corrected.
  • It would be better to give up to date references. Thanks for pointing this out. We agree some of the references we included are old. Those are the only articles that describe isoniazid pharmacokinetics in infants and toddlers in low TB burden countries until the current study performed in Spain. We still think these need to be cited, to emphasize the lack of literature investigating the pharmacokinetics of antiTB drugs in young children in this setting. As opposite to this, we have also included the most recent references of the papers from high TB countries, all of which have been published in the last decade.
  • Keywords: I would like to recommend the adhere MeSH headings. Thanks for the suggestion. We have made sure to use only MeSH headings as keywords in our paper (line 60).

Reviewer 3 Report

The first sentence describing prevalence and mortality due to TB should be supported by reference. similarly, certain lines, figures need adequate referencing.

Authors described recent prevalence or incidence of TB in Spain in second last paragraph, it should place above in the first paragraph to show that Spain is low TB burden country. 

Author Response

The first sentence describing prevalence and mortality due to TB should be supported by reference. similarly, certain lines, figures need adequate referencing. 

Thanks for pointing this out. Actually, the reference has been updated with the very last WHO TB Global Report in 2022 (line 63)

Authors described recent prevalence or incidence of TB in Spain in second last paragraph, it should place above in the first paragraph to show that Spain is low TB burden country.  

Thanks again for the suggestion. It makes sense to put together the sentences on epidemiology in the same para, now in line 67. References have been corrected accordingly.

Round 2

Reviewer 1 Report

The authors responded to all the raised comments by the reviewer